# *Staphylococcus haemolyticus* and *Providencia vermicola* Infections Occurring in Farmed Tilapia: Two Potentially Emerging Pathogens

**DOI:** 10.3390/ani13233715

**Published:** 2023-11-30

**Authors:** David Rajme-Manzur, Jorge Hernández-López, Marcel Martínez-Porchas, Francisco Vargas-Albores, Estefanía Garibay-Valdez, Daniel Eduardo Coronado-Molina, Miguel Ángel Hernández-Oñate, Francisco Vázquez-Ramírez, Luis Alfonso Velázquez-Valencia, Azucena Santacruz

**Affiliations:** 1Centro de Investigación en Alimentación y Desarrollo, A.C. Coordinación de Ciencia y Tecnología de Alimentos de Origen Animal, Biology of Aquatic Organisms, Hermosillo 83304, Sonora, Mexico; david.rajme17@estudiantes.ciad.mx (D.R.-M.); estefania.garibay@ciad.mx (E.G.-V.); miguel.hernandez@ciad.mx (M.Á.H.-O.); santacruz@ciad.mx (A.S.); 2Centro de Investigaciones Biológicas del Noroeste S. C. (CIBNOR), Unidad Hermosillo, Hermosillo 83106, Sonora, Mexico; jhlopez04@cibnor.mx (J.H.-L.); dcoronado04@cibnor.mx (D.E.C.-M.); 3Comité Estatal de Sanidad Acuícola de Chiapas A.C., Tuxtla Gutiérrez 29020, Chiapas, Mexico; gerencia.cesach@gmail.com (F.V.-R.); coordinacion.cesach@gmail.com (L.A.V.-V.)

**Keywords:** Nile tilapia, infectious diseases, *Staphylococcus*, *Providenciosis*, fish disease

## Abstract

**Simple Summary:**

The development of the production of freshwater species such as Nile tilapia has favored the increase in larger-scale infectious outbreaks and new diseases in this industry. This work aimed to determine the presence of bacterial diseases in Nile tilapia cultured in Chiapas, Mexico. Blood and viscera samples were taken from healthy and sick animals from several commercial farms. The isolated bacteria were identified and characterized by different laboratory tests. The isolation, identification and characterization of two pathogens involved in an infectious process were achieved: *Staphylococcus haemolyticus* and *Providencia vermicola*. Both bacteria were cataloged as causal agents of diseases in tilapia farming in Mexico. This is the first report of bacterial diseases in tilapia farms caused by two rare but potentially emerging pathogens for the species. These results make it possible to know the health situation of the region, estimate economic losses, know the behavior of pathogens and propose effective prevention and control measures.

**Abstract:**

This work aimed to determine the presence of bacterial pathogens in fish with a clinical picture suggestive of infectious disease in Nile tilapia reared in Chiapas, Mexico. Blood and viscera samples were taken from healthy and diseased animals from commercial farms. Clinical and pathological examinations of each individual were performed and samples were collected for bacteriological studies. The bacterial isolates were identified and characterized by culture, biochemical tests, antibiogram, challenge tests and 16S rRNA sequencing. *Staphylococcus haemolyticus* and *Providencia vermicola* were isolated from various diseased organisms. The clinical picture caused by *Staphylococcus haemolyticus* was characterized by appetite disorders, neurological signs, nodulation or ulceration in different areas and congestion or enlargement of internal organs. Providenciosis in juvenile specimens caused a characteristic picture of hemorrhagic septicemia. Challenge tests performed in healthy organisms revealed that both infections caused higher mortality rates in fish (*p* < 0.05) compared with non-infected specimens, with 100% survival. There was 100% mortality for animals infected with *P. vermicola* after three days post infection and 45% for those infected with *S. haemolyticus*. The isolation and identification of two pathogens involved in an infection process were achieved and cataloged as potential causal agents of disease outbreaks in tilapia farming in Mexico. This is the first report of possible bacterial infection caused by *S. haemolyticus* and *P. vermicola* in tilapia farms, which are two uncommon but potentially emerging pathogens for the species.

## 1. Introduction

Nile tilapia (*Oreochromis niloticus*) is farmed in different regions with diverse environmental conditions, establishing significant markets derived from its culture due to its adaptive capacity. The extensive Nile tilapia culture in Mexico is thriving [1,2] and produces several thousand tons per year [3,4]. However, bacterial diseases significantly harm the Nile tilapia aquaculture industry [5]. In commercial farms, mortality can reach 80% even in animals that have reached commercial sizes, with the consequent loss of significant investments already made in managing and feeding these crops.

The water quality, environmental conditions, and occurrence of pathogenic bacteria are determining factors for developing infectious diseases and optimal crop development [6,7]. Under stress conditions, fish become susceptible to infections produced by opportunist bacteria, including *Streptococcus agalactiae*, *S. iniae*, *Lactococcus garviae*, *Francisella noatunensis*, *Flavobacterium columnre*, *Staphylococcus* spp., *Aeromonas* spp. and *Providencia* spp. [8,9,10,11,12,13,14,15,16,17,18].

In general, Gram-negative bacteria such as *Aeromonas* spp., *Edwarsiella* spp. and *Providencia* spp. cause bacterial hemorrhagic sepsis syndrome [14]. Fish affected by this syndrome show signs of darkening of the skin, exophthalmia, anorexia and hemorrhagic or ulcerated areas on the bases of the pectoral and ventral fins and in the eye region. SHB can manifest itself and cause losses of 60–100% in farmed tilapia [19]. Gram-positive bacteria such as *Nocardia* spp., *Streptococcus* spp. and *Staphylococcus* spp. often cause chronic problems and granulomatosis. Affected tilapia may show disoriented and erratic swimming movements, ulcers and nodules on the body surface, unilateral or bilateral exophthalmia with or without corneal opacity and periocular hemorrhage [15,19].

The intensification of production conditions in the aquaculture industry has resulted in the occurrence of new infectious diseases caused by strains with a higher degree of virulence and lethality. Emerging bacterial diseases have been reported in Nile tilapia, such as franciselosis caused by *Francisella*, streptococcosis caused by *Streptococcus agalactiae* serotype IX, hemorrhagic septicemia by *A. veronii* and *A. jandae*, edwardsielosis caused by *Edwardsiella ictaluri* and other emerging infections caused by *Aerococcus viridans* [14] and other bacteria.

Furthermore, two or more bacteria have been frequently identified as causing simultaneous infectious processes or coinfections [20,21,22,23]. The early diagnosis of these processes and the identification of the causal agents that affect aquaculture in some geographical regions are vital for the timely application of control and preventive measures. Additionally, detecting unexpected emerging pathogens in regions susceptible to epidemics is crucial for the wealth performance of aquaculture industries.

The present work aimed to isolate, identify and confirm potentially emerging bacterial pathogens in commercial Nile tilapia farms from Chiapas, Mexico, by sampling fattening category organisms.

## 2. Materials and Methods

### 2.1. Sampling Animals

Sampling was conducted in several Nile tilapia farms in Chiapas, Mexico (see Appendix B), reporting disease symptoms. Four farms were visited: “Campo Viejo” farm located in La Angostura Dam, the largest freshwater reservoir in Mexico. The three other farms with an approximate distance of 30 km between each one of them (“Agua Blanca”, “Cañada el Dorado”, “La Cima”) were located in the Malpaso Dam. Overall, the water quality parameters of these farms registered similar values, including temperature: 28.5 ± 1.0 °C, dissolved oxygen: 6.57 ± 0.30 mg/L, pH: 7.5–7.8, total ammonia nitrogen: 0.06 ± 0.02 mg/L and nitrites: 0.01 ± 0.005 mg/L. Nile tilapia culture in these farms is based on an intensive production system of floating cages on the reservoir with a density of 80–120 animals/m^3^.

Twenty fish from the fattening categories (adults, 250–500 g) were sampled from each farm. Thirty of them showing signs and external lesions suggestive of bacterial infectious disease and fifty clinically healthy specimens were obtained from the four farms. The selected fish were sacrificed according to the Aquatic Animal Health Code of the World Organization for Animal Health [24].

Before dissection, lesions or other superficial signs were located and recorded on each specimen collected. Based on the information obtained from the clinical and pathological examination of the sampled animals, the proportion of the presentation of external and internal signs and lesions was determined. The signs and lesions of the sampled fish were considered low (30%), moderate (30% and 60%) and high (>60%) frequency, according to the proportions of the different signs and lesions corresponding to the number of times that they occurred.

### 2.2. Bacteriological Analysis and Phenotypic Characterization of Isolates

Samples were collected aseptically from skin lesions and internal organs (liver, spleen, heart). The tissue fragments were incubated at 37 °C for 24 h in Soya Trypticase Broth (STB) and swab samples in Blood Base Agar (BBA) and Soya Trypticase Agar (STA). Subsequently, Gram stains and optical microscope observations were performed [25]. After that, seeding was carried out in selective culture media. McConkey Agar (Oxoid) and Pseudomonas Agar (Oxoid) were used to isolate Gram (−) microorganisms. Likewise, to obtain pure cultures of Gram (+) microorganisms, seeding in Salt Mannitol Agar (SMA) and BBA added with antimicrobials (Oxoid): Oxolinic acid (5 mg/L) and Nalidixic acid (10 mg/L) was used as a strategy.

Different biochemical tests (catalase and oxidase, motility, esculin and hemolysis) were carried out for the pure bacterial cultures obtained. In addition, in the cultures of Gram (−) microorganisms, the API 20E system (BioMerieux, Vitek, Hazelwood, MO, USA) was used according to the manufacturer’s instructions.

The antimicrobial susceptibility of the bacterial isolates was determined using the disk diffusion method on Mueller–Hinton Agar as recommended by the Clinical and Laboratory Standards Institute. The resistance (R)/sensitivity (S) patterns of two Gram (+) and two Gram (−) isolates were analyzed.

### 2.3. Molecular Identification of Bacteria

Bacterial isolates were identified from 16S rRNA sequencing. Initially, genomic DNA was extracted from two colonies of each isolate (one Gram [+] and one Gram [−]) using DNAzol reagent (Invitrogen, San Diego, CA, USA) according to the manufacturer’s instructions. Amplicons of the complete 16S rRNA gene were obtained by PCR using primers F8 Fw: AGAGTTTGATCMTGGCTC and 1492R Rv: GNTAC-CTTGTTACGACTT (T4 Oligo) and the Platinum II Hot-Start PCR Master Mix kit (2X) (Invitrogen TM) following the manufacturer’s recommendations. The reaction was carried out in a 20 uL volume in a LongGene thermocycler under the following conditions: stage 1 (initial denaturation): 3 min at 95 °C; stage 2: 95, 60 and 72 °C for 30 s, respectively (repeated in 35 cycles); stage 3 (final extension) 72 °C for 5 min. To observe the products of the 16S rRNA amplification, electrophoresis was run on agarose gel at 1% (*w*/*v*) in 1X TAE stained with GelRed. The gels were visualized on a Gel DocTM XR+ transilluminator (Bio Rad, Hercules, CA, USA).

Amplicons of the complete 16S rRNA gene were bidirectionally sequenced by the Sanger method. The sequences obtained were compared with the GenBank database using the BLAST tool (Basic Local Food Search Tool) from the NCBI database, optimizing for highly similar sequences.

In addition, the 16S rRNA sequences from bacteria taxonomically related to the microorganisms putatively identified by BLASTN were obtained from GenBank. An alignment with Clustal W was performed and phylogenetic trees were constructed by Maximum Likelihood (ML) methods [26]. Phylogenetic reconstructions were performed with ETE3 3.1.2 [27] implemented on GenomeNet (https://www.genome.jp/tools/ete/, accessed on 1 October 2023). The ML tree was inferred using PhyML v20160115 executed with model and parameters: --pinv e --alpha e --nclasses 4 -o tlr -fm --bootstrap -2 [28].

### 2.4. Virulence Assay

A challenge test was performed to evaluate the pathogenic potential of the identified bacterial isolates. Gram (+) and Gram (−) isolates from diseased organisms were tested to determine their ability to cause disease in apparently healthy animals.

Three replicates of the bioassay were simultaneously carried out, and each replicate consisted of twenty four fish weighing 25 ± 5 g obtained from a commercial farm with no recent records of diseases (e.g., eight individuals were considered in each group). The fish were randomly distributed in each replicate into two experimental groups and one control.

Challenges were carried out using bacterial inoculum prepared from 24 h cultures. A specific infective dose of each bacterium was used for the treatments as follows: Gram (+) bacteria 1 × 10^10^ CFU/mL and Gram (-) bacteria 1 × 10^8^ CFU/mL based on previous tests. In all cases, the fish were inoculated intraperitoneally (IP) using a volume of 0.2 mL. The control group received an IP injection of 0.2 mL of 0.9% saline solution.

Clinical signs were recorded daily and dead fish were removed. Fish involved in this experiment were handled following the legislation for the welfare of experimental animals, Official Mexican Standard-062-Z00-1999 (NOM-062-Z00-1999) [29], on the Technical Specifications for Production, Care and Use of Laboratory Animals; all living fish were euthanized by anesthesia overdose (250 mg/L MS-222), a protocol authorized by the internal CIAD Ethics Committee (Registration: CONBIOÉTICA-26-CEI-001-20200122).

Finally, the presence of both bacteria was confirmed from organ samples (spleen, liver) and blood from dying fish. Sowing in general and selective culture media, bacterial DNA extraction, endpoint PCR amplification and 16S rRNA gene sequencing were carried out as previously described.

### 2.5. Histopathological Examination

To identify possible tissue changes caused by bacterial infection, a histopathological analysis of the spleen of some of the challenged fish (infected and non-infected) was performed. This organ was selected since it plays a central role in the immune response. The collected fragments were fixed with 10% neutral buffered formaldehyde, dehydrated with ethanol and then embedded in paraffin. Paraffin blocks were sectioned and stained with hematoxylin and eosin. The samples were then examined under an optical microscope.

## 3. Results

### 3.1. Clinical-Lesional Picture of the Sampled Organisms

During the study, moderate mortality (about 15%) was registered in three farms (Campo Viejo, Cañada el Dorado, La Cima) while detecting a clinical picture with a tendency to chronicity. However, high acute mortality (above 50%) was recorded on one farm (Agua Blanca).

In general, the most frequently observed clinical lesional picture was characterized by appetite disorders, neurological signs, nodulation or ulceration in different areas of the external surface of the fish and congestion or enlargement of internal organs (Figure 1; Appendix B). In a lesser proportion, fish with skin hyperpigmentation, deep muscle fibrosis, liver nodules and hemorrhagic lesions in internal organs were observed.

The predominant clinical signs of fish reared in the fattening category were of neurological origin. Behavior disorders and abnormal swimming were observed, including erratic swimming (in circles), stiffness and dorsal curvature, lethargy, food rejection and environmental indifference (Table 1). Likewise, there was a remarkable presentation of external lesions, mainly hemorrhagic ulcers in the head and gill region. The main pathological lesions were hepatosplenomegaly, petechial hemorrhages in the kidney and intestine, and ascites.

### 3.2. Bacteriological Analysis and Phenotypic Characterization of Isolates

Of the 30 clinically diseased fish sampled for bacteriological analysis, nine bacterial growths were obtained in BBA and TSA at 24 h of incubation. Three Gram (+) isolates were obtained using Staphylococcus selective culture media (SMA), generally observed as spherical or ovoid cells grouped in pairs, chains and clusters. Growth of gray colonies and β hemolysis was detected in BBA with antibiotics. In addition, the catalase test was positive in all cases. On the other hand, these bacteria did not grow on Pseudomonas or McConkey Agars. The integration of these elements determined Staphylococcus spp. It is important to note that isolates of this bacterium came from diseased fish from three of the four sampled farms: Campo Viejo, Cañada el Dorado and La Cima.

The other six bacterial isolates were observed as short Gram (−) rods, all growing in a selective medium for Pseudomonas Aeromonas. In this case, moderate growth occurred in MacConkey Agar in five isolates. Considering the results of the API 20 E (oxidase [+], Urea [−], Indole [+], citrate [+], arginine decarboxylase [−], gelatin [−], lysine decarboxylase [−], SH2 production [−], Voges-Proskauer [−]), these last five isolates were identified as *Providencia* spp. (Appendix A). The remaining isolate was considered to be a contaminating microbiota. Of the five *Providencia* spp. Isolates, four came from the Agua Blanca farm and only one from the Cañada el Dorado farm.

The isolates corresponding to *Staphylococcus* spp. were resistant to Oxolinic acid, Nalidixic acid, Ofloxacin, Ciprofloxacin, Erythromycin and Oxytetracycline; however, they were sensitive to Enrofloxacin, Fosfomycin, Chloramphenicol, Florfenicol, Cotrimoxazole and Penicillin G. The *Providencia* spp. isolates showed a marked sensitivity to Oxolinic acid, Nalidixic acid, Ofloxacin, Ciprofloxacin, Enrofloxacin, Fosfomycin, Florfenicol and Cotrimoxazole, and resistance to Erythromycin, Chloramphenicol, Penicillin G and Oxytetracycline. Resistance to oxytetracycline and erythromycin was detected for all the bacterial isolates evaluated (Table 2, Appendix A).

### 3.3. Molecular Identification Based on 16S rRNA

One Gram (+) pathogen and Gram (–) pathogens (presumptive *Staphylococcus* spp. and *Providencia* spp., respectively) were selected for the molecular identification of the bacterial isolates obtained. PCR amplification of the 16S rRNA gene of both bacteria showed a band of approximately 1300 bp (Figure 2).

When comparing the sequences obtained with those reported in the GenBank database, the maximum identity with *Providencia vermicola* (accession number NR_042415.1) was obtained for isolates corresponding to the genus Providencia spp. The Gram (+) isolates revealed the highest similarity with *Staphylococcus haemolyticus* (accession number NR_113345.1). The phylogenetic analysis showed that the isolates are grouped in the same clade as the bacteria for which they showed the highest similarity (Figure 3; Appendix C).

### 3.4. Virulence Assay

Both infections caused higher mortality rates in fish (*p* < 0.05) compared with non-infected specimens with 100% survival. During the trial, there were no deaths or alterations in the general condition of the fish in the control group.

The animals infected with *Providencia vermicola* showed a severe mortality rate of ≥70% in the first 24 h post infection and registered 100% mortality on the third day post infection (Figure 4). The infected fish presented lethargy and severe congestion in the gills and head region. On the other hand, the group infected with *Staphylococcus haemolyticus* showed less pronounced mortality, with 20% of fish dying during the first 24 h; after that, mortality continuously increased until the fourth day post infection (45%), but without registering more deaths from this point until the end of the challenge. Infected fish exhibited decreased activity (slower swimming), anorexia, torpor and darkening of the body; however, the fish that survived the entire bioassay did not show signs of disease.

Isolates of *Staphylococcus haemolyticus* and *Providencia vermicola* were recovered on TSA and BBA plates from blood, liver and spleen samples of necropsied animals. Isolates were confirmed by plating on selective media, biochemical tests and 16S rRNA amplification and sequencing (Figure 2). In the control group of fish, no bacterial growth was obtained in any culture media where blood and organ samples from these fish were seeded.

### 3.5. Histopathological Analysis

To determine the histopathological changes caused by infection by *S. haemolyticus* and *P. vermicola*, sections of spleen tissue were analyzed. In both cases, signs of hyperreactivity of the white pulp were observed, with an increase in the number of macrophages and melanomacrophage centers (MMC). Furthermore, in the spleen of fish infected with *S. haemolyticus*, areas of diffuse fibroblastic reaction of the stroma, hemosiderosis and accumulation of melanin pigments were identified. In uninfected animals, no alterations were observed in the tissue sections analyzed. (Figure 5).

## 4. Discussion

Bacterial diseases cause costly problems for Nile tilapia producers and other warm water fish. Outbreaks caused by bacterial pathogens can develop rapidly, causing high mortality rates (acute diseases) or, on the other hand, slow development and less severity but persistent for extended periods (chronic diseases) [30,31].

In the present study, the isolation of bacterial pathogens responsible for morbidity and mortality in commercial Nile tilapia farms in Chiapas, Mexico, was carried out. In general, there was moderate mortality and a certain tendency to chronicity on farms where *Staphylococcus haemolyticus* was isolated.

The predominant pathological picture observed in the fish where *Staphylococcus haemolyticus* was isolated included lethargy, anorexia and erratic swimming. Nodulation and ulceration were also observed in different areas of the external surface of the fish, together with congestion or enlargement of internal organs. Huang et al. [15] described the epidemiology and pathogenicity of *Staphylococcus epidermidis* in Nile tilapia (*Oreochromis* spp.) in Taiwan. Sick fish showed splenomegaly with the spread of multiple white nodules and lesions in the spleen and kidney. In *Staphylococcus epidermidis* infections, apoptotic cells were observed mainly in lymphocytes and macrophages in the spleen, kidney and occasionally in the brain, liver, gonads, mesentery, stomach, intestine and skeletal muscle [32].

Staphylococci species occur as commensal colonizers of the skin and mucous membranes of different species of animals, including humans, but they are not part of the fish microbiota [33]. In fish, infections by *Staphylococcus* sp. generally cause hemorrhagic septicemia, affecting many species of productive interest, such as sparids, salmonids, tilapia species and carp [34].

Until now, there have been few reports of *Staphylococcus* infections in fish and they have documented a low epidemiological impact (generally sporadic outbreaks) [35]. For example, Kanchan et al. [36] obtained a 20% prevalence of *S. epidermidis* in *Anabas testudineus.* However, in China, considerable mortality levels have been found frequently due to this disease. In finless eels (*Monopterus albus*) infected by *Staphylococcus*, the main signs were swelling of the head and anus and wound suppuration. Similar signs caused by *Staphylococcus* were also detected in the golden arowana (*Scleropages formosus*) [37]. In addition, in zebrafish, there was a clear sign of the disease after experimental infection by *S. aureus*, such as gill congestion and hemorrhage, intestinal ulcers, hepatorrhagia and milky eyeball [37].

On the other hand, in the Agua Blanca farm (where *Providencia vermicola* was isolated) high acute mortality was recorded. Injuries were represented by signs suggestive of a septicemic process. Fish mainly presented anorexia, lethargy or moribund, with abnormal swimming and a distended abdomen. Likewise, there was a more striking presentation of external lesions in this group of animals, mainly hemorrhagic ulcers in the region of the head and gills. The main pathological lesions were hepatosplenomegaly, petechial hemorrhages in the kidney and intestine, and ascites.

Regarding the macroscopic pathological lesions, the Nile tilapia specimens from which *P. vermicola* was isolated showed signs like those described by other authors [18,38], including the presence of ulcers, serosanguineous ascitic fluid, adhesions, liver color and irregular appearance, enlarged spleen and hemorrhages. Moreover, ornamental fish infected with *P. vermicola* and other enterobacteria showed lethargy, anorexia, abnormal swimming, hemorrhages, scale discoloration and gill necrosis [39].

*Providencia* spp. is found mainly in aquatic environments and involved in infections of birds and mammals. The species *P. rettgeri* and *P. vermicola* have been isolated from diseased fish [17,18,40,41]. Few reports on bacterial diseases have documented *Providencia* spp. as an emergent etiologic agent of the hemorrhagic septicemia complex in freshwater species [37]. Some positive diagnoses for this bacteriosis, caused explicitly by *Providencia rettgeri*, showed the economic impact of the pathogen in aquaculture due to high mortality and growth decrease in infected animals [17,42]. Mortality in fish infected with *Providencia* spp. is catalyzed by inadequate handling, stress due to transport and poor water quality [43]. According to Bejerano et al. (1979), in silver carp (*Hypophthalmichthys molitrix*) the disease is characterized by red ulcers on the abdomen, pectoral fin and around the head. Other pathologies include orbital edema, hemorrhage and focal necrosis of the sclera.

The challenge test to evaluate the pathogenic potential of the isolates confirmed that both bacteria could produce significant mortality rates in fish. *Providencia vermicola* produced a higher mortality rate than in the group challenged with *Staphylococcus haemolyticus*. Likewise, deaths occurred gradually in the group challenged with this last bacterium. This shows that *P. vermicola* could have a greater pathogenic capacity and a faster infection rate [44], representing a potential threat to Nile tilapia farms.

Despite the fact that both isolated bacteria are not common in Nile tilapia aquaculture, at least not as pathogens, the laboratory challenges corroborated their virulence and aggressiveness. The experimental infection of Nile tilapia with *P. vermicola* suggested a clinical picture compatible with hemorrhagic septicemia since there were evident signs and alterations of clinical pathology. Signs included anorexia, lethargy, increased opercular frequency and surface gasping, which were also found in natural infections caused by *Providencia rettgeri* in Nile tilapia [44]. The necropsy findings were similar to those reported by Souza et al. in *P. rettgeri* infections: systemic and local hemodynamic changes such as congestion, hemorrhages, increased vascular permeability, ascites and expulsion of bloody fluid through the anus and the opercula. Similarly, in goldfish infected with *P. vermicola*, renal and hepatic congestion, splenomegaly and ascites were observed at necropsy [39].

On the other hand, the animals challenged with *Staphylococcus haemolyticus* showed less dramatic clinical signs and pathological lesions. Similarly, rainbow trout (*Oncorhynchus mykiss*) challenged with *Staphylococcus xylosus* presented low mortality and exophthalmia [45]. However, it was found that this bacterium also has the potential to cause disease and fish death. Therefore, the experimental approach revealed that both *P. vermicola* and *S. haemolyticus* previously isolated from the farmed Nile tilapia resulted in a clinically aggressive pathogen to Nile tilapia, placing them as potential emerging pathogens for the species.

It is important to emphasize that the 16S rDNA sequences of the bacterial isolates turned out to be identical to those obtained from the bacteria recovered from the experimentally challenged fish. This finding confirms that the bacterial isolates used are responsible for the disease and mortality obtained in the experimental challenge. The isolates used in the challenge test were recovered as pure cultures in all the diseased fish that were sampled, but no growth was obtained in the control group samples, where they remained clinically healthy. In this way, Koch’s postulates, which are designed to establish the causal relationship between a pathogenic biological agent and a specific disease, are fulfilled.

Bacterial infections have increased considerably with the intensification of aquaculture and are responsible for significant economic losses in this industry worldwide [46,47]. Diseases caused by these pathogens are among the most common and probably the most critical problems of freshwater species. Notably, this is the first study focused on identifying and characterizing such bacterial pathogens as causative disease agents in commercial Nile tilapia culture in the region with the highest production in Mexico. The occurrence of harmless bacteria becoming pathogenic is not unusual in aquaculture, and recent evidence has demonstrated that harmless or commensal bacteria can mutate at such a rate that they can adapt to the environment. Thus, when the environment of such bacteria is the immune response of a host, the use of antibiotics or other adverse scenarios, bacteria mutate from harmless to life-threatening pathogens.

This report constitutes the first isolation of *Staphylococcus haemolyticus* and *P. vermicola* from clinically diseased Nile tilapia on commercial farms in Mexico. The findings described so far should be considered in future epidemiological studies and the diagnostic approach of outbreaks with suspicion of infectious diseases of bacterial origin.

## 5. Conclusions

Although *Staphylococcus haemolyticus* and *Providencia vermicola* bacteria are not common pathogens in Nile tilapia aquaculture, the results show that they are not only present in farms but also have a high virulence, causing higher mortalities than expected in a fish like Nile tilapia that is usually resistant to diseases, representing a risk to the activity. Therefore, both species should be considered an emerging concern in fish farming.

## Figures and Tables

**Figure 1 animals-13-03715-f001:**
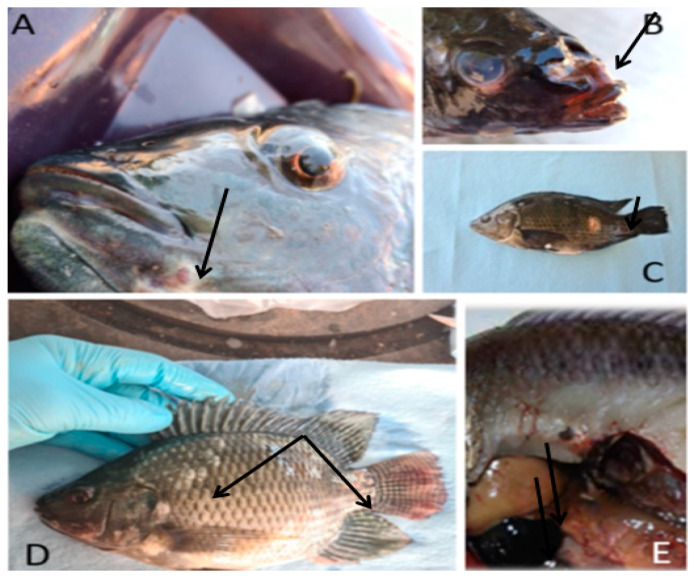
External lesions in Nile tilapia (*Oreochromis niloticus*) affected by an infectious disease. (**A**): nodules in the mandibular region, (**B**): ulceration in the head region, (**C**): ulcer on the lateral body surface, (**D**): loss of scales, (**E**): gallbladder distention and liver congestion.

**Figure 2 animals-13-03715-f002:**
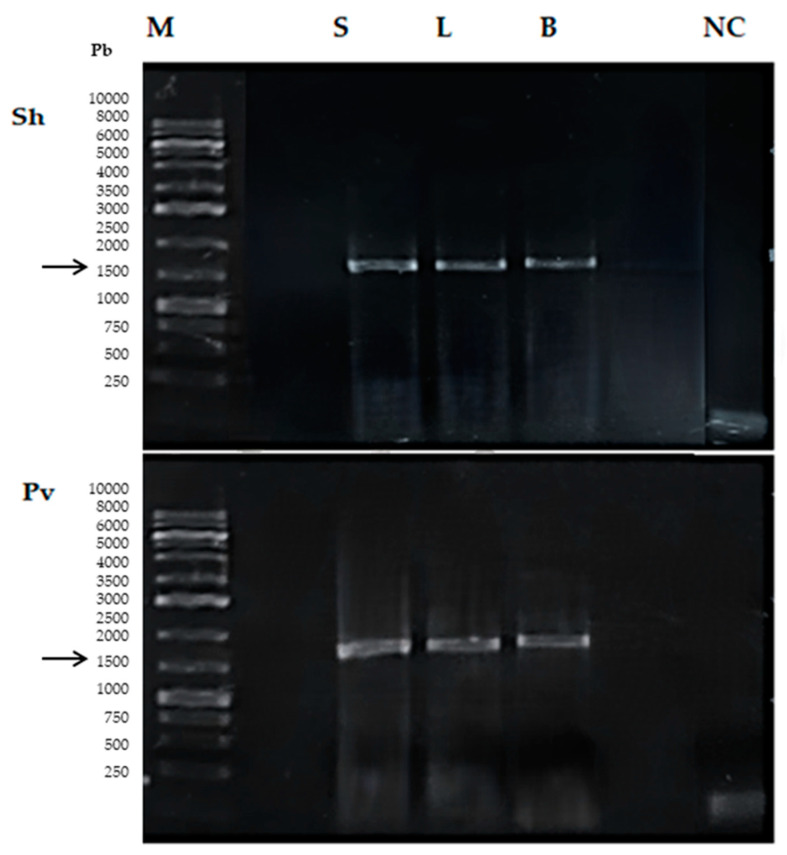
Amplicons of the 16S rRNA gene obtained by endpoint PCR. M, molecular marker (16S rRNA gene). PCR products are shown from DNA extracted from bacterial colonies from spleen (S), liver (L) and blood (B) samples (cultured on BBA) of fish challenged with *S. haemolyticus* (Sh) and *P. vermicola* (Pv) NC: negative control (DNase-free distilled water). The black arrow indicates the amplicon size (~1300 bp).

**Figure 3 animals-13-03715-f003:**
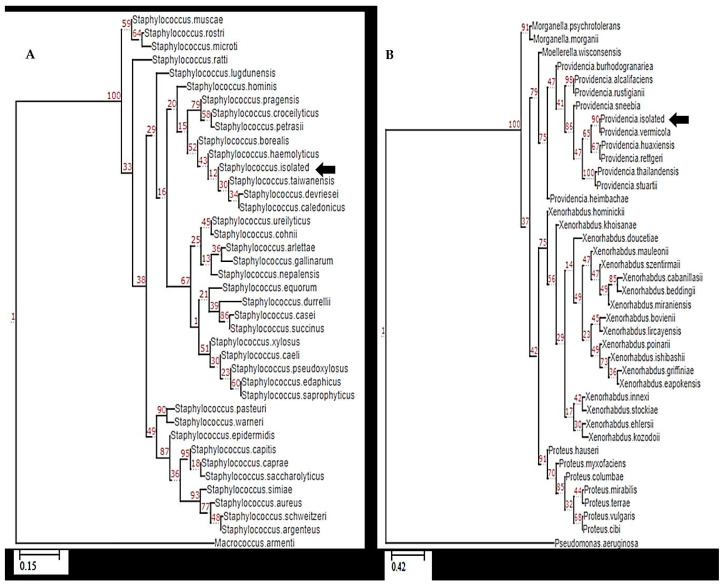
Phylogenetic relationship between Mexican isolates (black arrow) and other bacterial species, according to 16S rRNA gene sequence analysis. The Gram (+) isolate was phylogenetically located in the same group as *S. haemolyticus* (**A**). The Gram (−) isolate was located in the same group of *P. vermicola*, very close to *P. rettgeri* (**B**). The tree was rooted using Macrococcus armenti (**A**) and Pseudomonas aeruginosa (**B**) as outgroups. Note: The branches are labeled with parametric values based on Chi2 returned by the approximate likelihood ratio test, and the names of the nodes correspond to the aligned sequences. The scale bar represents nucleotide change per nucleotide position.

**Figure 4 animals-13-03715-f004:**
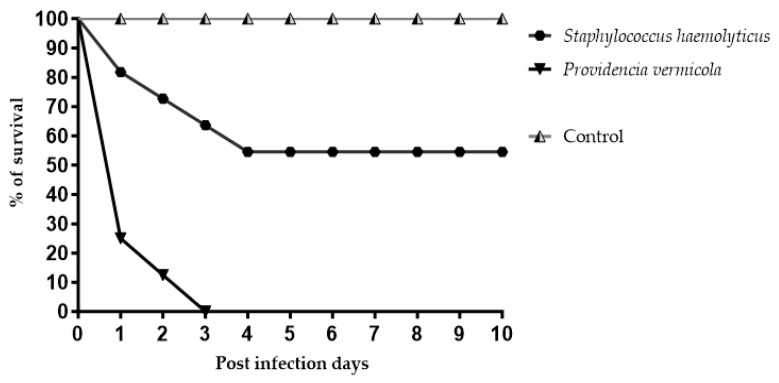
Survival rate of fish challenged in the laboratory with isolates of *Providencia vermicola* and *Staphylococcus haemolyticus* obtained from sampled commercial farms. The average values of daily survival obtained from each group from the three repetitions of the challenge are shown.

**Figure 5 animals-13-03715-f005:**
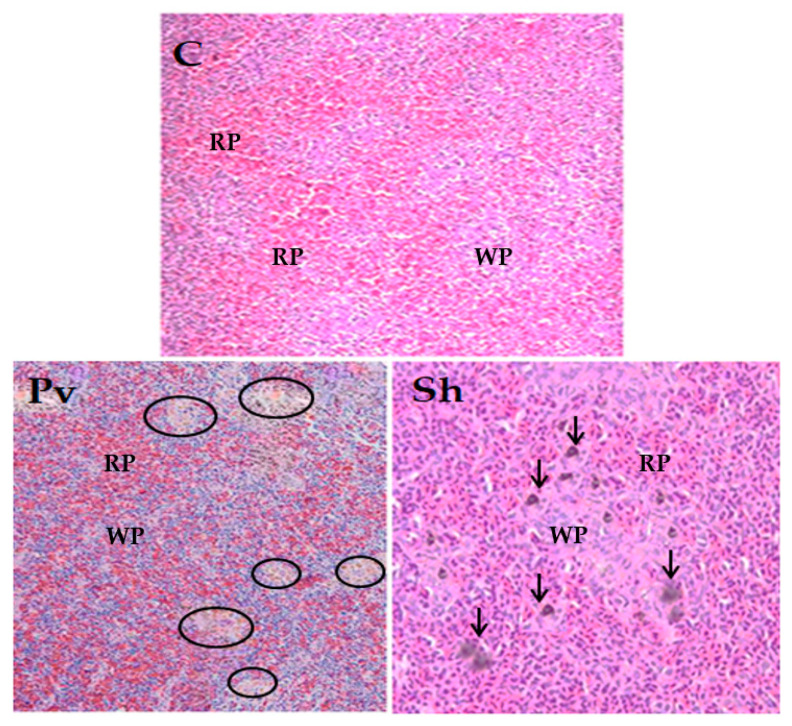
Histopathological analysis of the spleen of fish included in the virulence assay. C: uninfected fish (control group), tissue without alterations; Pv: fish infected with *P. vermicola*; hyperreactivity of the white pulp (WP) and decrease in the density of the red pulp (RP) of the spleen with MMC proliferation (black circle) are observed; Sh: fish infected with *S. haemolyticus*. The black arrows point to the MMC proliferation and melanin accumulation.

**Table 1 animals-13-03715-t001:** Main signs and lesions detected in diseased fish sampled from commercial farms. Absent alteration (-), low frequency (+), moderate frequency (++), high frequency (+++). The signs and lesions of the sampled fish were considered low (30%), moderate (30% and 60%) and high (>60%) frequency, according to the proportions of the different signs and lesions corresponding to the number of times that occurred.

Signs/Injuries	Presentation Frequency
Reduced food intake	+++
Erratic swimming	+++
Distended abdomen	+++
Ulcers in the head	+++
Eye injuries	-+
Ascites	++
Friable liver	++
Splenomegaly	+
Hemorrhage in internal organs	++
Mortality	++

**Table 2 animals-13-03715-t002:** Antimicrobial susceptibility of two isolates of *Staphylococcus* spp. and *Providencia* spp. obtained from diseased organisms in the evaluated farms.

Antimicrobial	Bacterial Genus			
	*Staphylococcus*	*Staphylococcus*	*Providencia*	*Providencia*
Oxolinic acid	R	R	S	S
Nalidixic acid	R	R	S	S
Ofloxacin	R	R	S	S
Ciprofloxacin	R	S	S	S
Enrofloxacin	S	S	S	S
Erythromycin	R	R	R	R
Fosfomycin	S	S	S	R
Chloramphenicol	S	S	R	S
Florfenicol	S	S	S	S
Oxytetracycline	R	R	R	R
Cotrimoxazole	S	S	S	S
Penicillin G	S	S	R	R

S: sensitive, R: resistant.

## Data Availability

The data presented in this study are available on request from the corresponding author.

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
