# Peer review of "Staphylococcus haemolyticus and Providencia vermicola Infections Occurring in Farmed Tilapia: Two Potentially Emerging Pathogens"

_animals, 2023, doi:10.3390/ani13233715_

Round 1

Reviewer 1 Report

Comments and Suggestions for Authors

This study examined possible outbreaks of infection in several tilapia farms. The study identified two bacterial species as possible new pathogens. Experimental data is provided to demonstrate the disease-causing potential of both bacterial species. The study is really well done and presented. Small technical issue, it would be nice to have the figures and tables closer to where they are first cited. 

In the abstract a couple of hyphens have gotten in where they shouldn’t be. The bacterial species should be in italics.

Lines 34 and 37: remove hyphens from extensive and consequent.

Lines 45-61: the reference style is inconsistent with the rest of the manuscript.

Line 158: remove hyphen in standard.

Line 162: remove 3. And the send.

Line 279: replace ‘where’ with ‘with’.

Line 327: corroborated and revealed seems redundant.

Line 352: remove hyphen in responsible.

Lines 356-361: rewrite the sentence as at least two.

Author Response

his study examined possible outbreaks of infection in several tilapia farms. The study identified two bacterial species as possible new pathogens. Experimental data is provided to demonstrate the disease-causing potential of both bacterial species. The study is really well done and presented.

-Thank you for the comments, which contributed to improving the overall quality of the manuscript.

Small technical issue, it would be nice to have the figures and tables closer to where they are first cited. 

-Figures and tables have been placed closer to where they are first cited to make the text easier to understand

In the abstract a couple of hyphens have gotten in where they shouldn’t be. The bacterial species should be in italics.

-Incorrectly placed hyphens in the abstract and other sections of the article have been removed, and bacterial names that were not italicized have been corrected

Lines 34 and 37: remove hyphens from extensive and consequent.

-Corrected

Lines 45-61: the reference style is inconsistent with the rest of the manuscript.

-Corrected

Line 158: remove hyphen in standard.

-Corrected

Line 162: remove 3. And the send.

-Corrected

Line 279: replace ‘where’ with ‘with’.

-Corrected

Line 327: corroborated and revealed seems redundant.

-Corrected

Line 352: remove hyphen in responsible.

-Corrected

Lines 356-361: rewrite the sentence as at least two.

-Corrected

Reviewer 2 Report

Comments and Suggestions for Authors

I was honored to review the manuscript entitled “Staphylococcus haemolyticus and Providencia vermicola infections occurring in farmed tilapia: two potentially emerging pathogens” submitted to Animals.

I would like to thank authors for preparing this manuscript. The number of samples used in this study are fine. Herein, the reviewer thought that this manuscript needs to improve in structure. The structure of this paper should be corrected for preparing the readable version. We have a lot of methods that you can use for confirming the results such as IHC, histology and PCR methods for organs. Finally, there are some points that the authors should correct them and still needs some change for improving the manuscript.

There are some points to correct:

1.     Line 20, 21, 27,59,188,197,198,200,203,210: bacterial species should be written in italic.

2.     Line 34: ex-tensive? What is the meaning? Line 37: con-sequent? Line170: dis-orders? Line352: re-spensible?

3.  As I know “organisms” is used for bacterial, viral and parasitical species not use for animal species. Please replace this word with appropriate word in the whole text.

4. Line 46,49,54,58,61,139,153,299,332: References should be written according to guideline of journal.

5.  Line 57: Have been reported

6.  Line 162: what is the 3 at the end of sentence?

7.  The authors should prepare an accession number for these species.

8.  Prepare an image of API test and Antibiogram test in the supplementary material.

9. I am wondering that why you did not use these organs for PCR and confirm the results with culture. Please provide the section in material and methods and analyze the organs with PCR.

1.  Please provide a histology part to improve the quality of the manuscript.

1.  Please provide some data in discussion to support your manuscript.

In conclusion, this manuscript can be published after adding these supporting methods and major revision.  

Author Response

I was honored to review the manuscript entitled “Staphylococcus haemolyticus and Providencia vermicola infections occurring in farmed tilapia: two potentially emerging pathogens” submitted to Animals.

I would like to thank authors for preparing this manuscript. The number of samples used in this study are fine. Herein, the reviewer thought that this manuscript needs to improve in structure. The structure of this paper should be corrected for preparing the readable version. We have a lot of methods that you can use for confirming the results such as IHC, histology and PCR methods for organs. Finally, there are some points that the authors should correct them and still needs some change for improving the manuscript.

-Thank you for your comments and suggestions, which contributed to improving the overall quality of the manuscript

There are some points to correct:

Line 20, 21, 27,59,188,197,198,200,203,210: bacterial species should be written in italic.

-Corrected

Line 34: ex-tensive? What is the meaning? Line 37: con-sequent? Line170: dis-orders? Line352: re-spensible?

     -Incorrectly placed hyphens have been removed

As I know “organisms” is used for bacterial, viral and parasitical species not use for animal species. Please replace this word with appropriate word in the whole text.

     -The word “organisms” that referred to fish was replaced throughout the document.

Line 46,49,54,58,61,139,153,299,332: References should be written according to guideline of journal.

     -References whose citation style did not match the journal's guidelines have also been corrected

Line 57: Have been reported

-Corrected

Line 162: what is the 3 at the end of sentence?

-Corrected

The authors should prepare an accession number for these species.

    -The accession number of the bacteria with which our isolates had 100% identity was included.

Prepare an image of API test and Antibiogram test in the supplementary material.

-An image of the biochemical tests in the API 20E miniaturized system and the antimicrobial susceptibility test for the two bacteria studied were included in supplementary material.

I am wondering that why you did not use these organs for PCR and confirm the results with culture. Please provide the section in material and methods and analyze the organs with PCR.

-Methods and an image of the electrophoretic run of the 16S rRNA gene amplicons obtained by endpoint PCR from organ and blood samples of the organisms sampled in the challenge assays were included as supplementary material.

Please provide a histology part to improve the quality of the manuscript.

-A section on histopathological analysis carried out with organ samples from the animals used in the challenge tests was added.

Please provide some data in the discussion to support your manuscript.

-In the discussion section, references and information considered relevant to support our results were added.

Reviewer 3 Report

Comments and Suggestions for Authors

Nile tilapia (Oreochromis niloticus) is farmed in different regions with diverse environmental conditions, establishing significant markets derived from its culture due to its adaptive capacity.However, bacterial diseases significantly harm the Nile tilapia aquaculture industry. In commercial farms, mortality can reach 80% even in organisms that have reached commercial sizes, with the con-sequent loss of significant investments already made in managing and feeding these crops. The present work aimed to isolate, identify, and confirm potentially emerging bacterial pathogens in commercial Nile tilapia farms from Chiapas, Mexico, by sampling fattening-category organisms.In general, the topic of the manuscript is very meaningful, but some issues need to be revised.

1. The abstract mainly lists the research results. The abstract need to be written.

2. There are many bacterial diseases in Nile tilapia,such as Streptococcus agalactiae, S. iniae. Staphylococcus haemolyticus and Providencia vermicola also can infect Nile tilapia, and both infections caused higher mortality rates in fish (P<0.05) compared with non-infected specimens with 100% survival. So I want to know which bacteria are the main pathogens that infect tilapia.

Comments on the Quality of English Language

Minor editing of English language required.

Author Response

Nile tilapia (Oreochromis niloticus) is farmed in different regions with diverse environmental conditions, establishing significant markets derived from its culture due to its adaptive capacity.However, bacterial diseases significantly harm the Nile tilapia aquaculture industry. In commercial farms, mortality can reach 80% even in organisms that have reached commercial sizes, with the con-sequent loss of significant investments already made in managing and feeding these crops. The present work aimed to isolate, identify, and confirm potentially emerging bacterial pathogens in commercial Nile tilapia farms from Chiapas, Mexico, by sampling fattening-category organisms.In general, the topic of the manuscript is very meaningful, but some issues need to be revised.

-Thank you for your comments and suggestions, which contributed to improving the overall quality of this manuscript.

  1. The abstract mainly lists the research results. The abstract need to be written.
  2. There are many bacterial diseases in Nile tilapia,such as Streptococcus agalactiae, S. iniae. Staphylococcus haemolyticus and Providencia vermicola also can infect Nile tilapia, and both infections caused higher mortality rates in fish (P<0.05) compared with non-infected specimens with 100% survival. So I want to know which bacteria are the main pathogens that infect tilapia.

-The summary was corrected according to your suggestions, and a simple summary  indicating such point was included.

Round 2

Reviewer 2 Report

Comments and Suggestions for Authors

I would like to thank authors for preparing the revise version of this manuscript. The present version is a good quality for publication and interest for readers.